# Effects of Ingesting Both Catechins and Chlorogenic Acids on Glucose, Incretin, and Insulin Sensitivity in Healthy Men: A Randomized, Double-Blinded, Placebo-Controlled Crossover Trial

**DOI:** 10.3390/nu14235063

**Published:** 2022-11-28

**Authors:** Aya Yanagimoto, Yuji Matsui, Tohru Yamaguchi, Masanobu Hibi, Shigeru Kobayashi, Noriko Osaki

**Affiliations:** 1Biological Science Laboratories, Kao Corporation, 2-1-3 Bunka, Sumida, Tokyo 131-8501, Japan; 2Health & Wellness Products Research Laboratories, Kao Corporation, 2-1-3 Bunka, Sumida, Tokyo 131-8501, Japan; 3Department of Surgery, Tokyo Rinkai Hospital, 1-4-2 Rinkai-cho, Edogawa, Tokyo 134-0086, Japan

**Keywords:** GIP, GLP-1, insulin sensitivity, type 2 diabetes, coffee chlorogenic acids, green tea catechins

## Abstract

Epidemiologic studies have revealed that consuming green tea or coffee reduces diabetes risk. We evaluated the effects of the combined consumption of green tea catechins and coffee chlorogenic acids (GTC+CCA) on postprandial glucose, the insulin incretin response, and insulin sensitivity. Eleven healthy men were recruited for this randomized, double-blinded, placebo-controlled crossover trial. The participants consumed a GTC+CCA-enriched beverage (620 mg GTC, 373 mg CCA, and 119 mg caffeine/day) for three weeks; the placebo beverages (PLA) contained no GTC or CCA (PLA: 0 mg GTC, 0 mg CCA, and 119 mg caffeine/day). Postprandial glucose, insulin, glucagon-like peptide-1 (GLP-1), and glucose-dependent insulinotropic polypeptide (GIP) responses were measured at baseline and after treatments. GTC+CCA consumption for three weeks showed a significant treatment-by-time interaction on glucose changes after the ingestion of high-fat and high-carbohydrate meals, however, it did not affect fasting glucose levels. Insulin sensitivity was enhanced by GCT+CCA compared with PLA. GTC+CCA consumption resulted in a significant increase in postprandial GLP-1 and a decrease in GIP compared to PLA. Consuming a combination of GTC and CCA for three weeks significantly improved postprandial glycemic control, GLP-1 response, and postprandial insulin sensitivity in healthy individuals and may be effective in preventing diabetes.

## 1. Introduction

Type 2 diabetes, a condition involving insulin resistance and impaired insulin secretion that progresses to a chronic hyperglycemic state [1], is a leading cause of death worldwide [1,2,3]. Ameliorating the sustained hyperglycemia resulting from whole-body insulin resistance is essential in preventing diabetes and its complications. Improving insulin resistance and controlling blood glucose by adopting healthy dietary habits and increasing physical activity in the early stages are essential factors for preventing diabetes onset; however, no effective non-pharmacologic methods for reducing insulin resistance have been established.

Blood glucose is strictly regulated by insulin, glucagon, and their feedback mechanisms. In addition, incretins, such as glucose-dependent insulinotropic polypeptide (GIP) and glucagon-like peptide-1 (GLP-1), are secreted from K cells in the upper part of the small intestine and from L cells in the lower part of the small intestine in response to dietary carbohydrates and fats, and they have important roles in regulating blood glucose levels via insulin secretion from pancreatic β-cells [4,5]. GLP-1 and GIP also have extrapancreatic effects and regulate physiologic functions related to glucose and energy homeostasis. The extrapancreatic effects of GLP-1 include increasing the glucose uptake capacity of the liver and muscle and improving whole-body insulin sensitivity [4,6], while extrahepatic GIP promotes fat accumulation via glucose and free fatty acid uptake in the adipocytes [7].

Green tea and coffee are popular beverages worldwide. Epidemiologic studies indicate that consuming green tea and coffee reduces type 2 diabetes risk. Individuals typically consuming over five cups of green tea [8] or three to six cups of coffee daily [9,10] are at a lower risk of type 2 diabetes. However, the effect of green tea or coffee on insulin resistance remains unclear [11,12,13,14], and the association of green tea and coffee with developing diabetes is also inconclusive. Green tea and coffee contain abundant polyphenols, such as green tea catechins (GTC) and coffee chlorogenic acids (CCA), respectively, which reduce the risk of developing type 2 diabetes and other metabolic diseases [15,16,17,18,19]. Venables et al. [20] reported that consuming 890 mg of green tea extract containing GTC increases insulin sensitivity by 13% in healthy individuals. Moreover, a single ingestion of CCA suppresses postprandial blood glucose levels [21] and promotes GLP-1 secretion [22]. Zuñiga et al. [23] reported that subjects with impaired glucose tolerance who consumed CCA (400 mg/d) for 12 weeks exhibited significantly reduced fasting blood glucose levels and a lower insulinogenic index. These reports suggest that due to their distinct mechanisms of action, the effects of GTC and CCA on glucose metabolism will be enhanced by their combined ingestion. To our knowledge, the effects of the combined consumption of GTC and CCA on hyperglycemia have not been evaluated.

Here, we hypothesized that the continuous combined consumption of GTC and CCA would improve the postprandial glycemic response in healthy men. We examined the effects of continuous combined consumption of GTC and CCA for three weeks on postprandial glucose, insulin and incretin responses, and insulin sensitivity in healthy men. The primary endpoint was the difference in the postprandial glucose response, secondary endpoints were changes in the insulin, GLP-1, and GIP responses; insulin sensitivity, evaluated using the Matsuda index; insulin resistance, evaluated using an index obtained by multiplying the glucose area under the curve (AUC) and the AUC of insulin over 2 h (AUC_ins×glu_); the homeostatic model assessment of insulin resistance index (HOMA-IR) and homeostasis model assessment β cell function (HOMA-β).

## 2. Materials and Methods

### 2.1. Subjects

Eleven healthy males (aged 20–60 years; fasting blood glucose ≤125 mg/dL) were recruited through emails on 26 September 2016, and our website described the study outline. Exclusion criteria were a history of diabetes or cardiovascular disease, hypertension, hypercholesterolemia, dyslipidemia, eating disorders, food allergies to the test food, excessive alcohol intake (≥30 g/day), smoking habit, being a shift worker, or being determined to be unqualified by the physician in charge. The study was conducted in accordance with the ethical principles of the Declaration of Helsinki and was approved by the ethics committee of the Kao Corporation (No. 16–35, 8 September 2016; Tokyo, Japan). All the participants received verbal and written explanations and provided written informed consent prior to registration. The trial was registered with the University Hospital Medical Information Network (UMIN; http://www.umin.ac.jp/ (10 November 2022); Registration No. UMIN000039514).

### 2.2. Study Design

In this randomized, double-blinded, placebo-controlled crossover trial including two 3-week intervention periods with a 2-week wash-out period, participants consumed either a GTC- and CCA-enriched beverage (GTC+CCA) or a placebo beverage (PLA) containing no GTC or CCA between September 2016 and December 2016 in the Tokyo metropolitan area, Japan. The participants were enrolled and randomly allocated to the starting condition with a random number generator by an independent researcher using the Excel spreadsheet program (Microsoft Excel 2010, Microsoft Corp., Redmond, WA, USA). All persons involved in the trial, including the participants, researchers who performed the interventions, and the supervising investigator, were blinded to the randomization. Participants were asked to maintain their usual dietary habits and physical activities, and their intake of green tea and coffee was limited to once daily during the entire study period. During the wash-out period, food intake and exercise were not guided or restricted. After a 1-week run-in period, baseline measurements were conducted after overnight fasting for at least 12 h before the participants consumed a test meal [24]. During the 3-week intervention periods, the participants consumed either the GTC+CCA or PLA beverages once daily after breakfast. The daily diets three days before each examination period were assessed by registered dietitians. The participants recorded individual dietary details of all meals during the three days using a diet-recording form to demonstrate that their eating habits and exercise levels had not changed. The content of the diet according to the dietary form and pictures was analyzed by a dietician using Excel-Eiyou-kun Ver. 6 software (Kenpakusha, Tokyo, Japan), and Hymena & Co (Tokyo, Japan) calculated the total calorie, carbohydrate, fat, and protein intake. A day before the measurements, the participants were prohibited from doing strenuous exercises, were required to ingest pre-packaged meals provided by the study coordinator (total energy content: 2412 kcal/day, 14 E% from protein, 25 E% from fat, and 61 E% from carbohydrate), and were asked to go to bed before midnight without ingesting any food or beverage other than water after 9:00 pm. The energy of the pre-packaged meals was 25 E% for breakfast, 24 E% for lunch, 21 E% for a snack, and 30 E% for dinner. The subjects arrived at the laboratory before 8:00 am on measurement day. Height, weight, body fat (body fat scale, model TF-780, Tanita Corp., Tokyo, Japan), waist circumference, and blood pressure (digital blood pressure monitor, model HEM-1000, Omron, Japan) were measured. Blood was drawn from a brachial vein under fasting conditions (0 h) and at 0.5, 1, 1.5, 2, 3, and 4 h after the test meal consumption (ingested within 15 min; meal test S, Saraya Co., Ltd., Osaka, Japan) and subsequent intake of the test beverage. The test beverages and meal test S were distributed by the study coordinator immediately prior to ingestion. The test meal was a high-fat cookie made from flour, butter, maltose, chicken egg, and baking powder, with an energy content of 592 kcal (6 E% from protein, 43 E% from fat, and 50 E% from carbohydrates). The outcome measurements were glucose, insulin, glucagon, C-peptide, GLP-1, and GIP concentrations; body weight; body fat percentage; and waist circumference. Furthermore, the exploratory measurements were for free fatty acids, ketone bodies, triglycerides, total bile acid, cortisol, glycoalbumin, 1,5-anhydroglucitol, high-sensitivity C-reactive protein, low-density lipoprotein cholesterol (LDL-C), high-density lipoprotein cholesterol (HDL-C), and total cholesterol.

### 2.3. Test Beverages

The following test beverages were prepared at the Kao Corporation: GTC+CCA enriched beverage containing 620 mg GTC, 373 mg CCA, and 119 mg caffeine; and PLA beverage containing 0 mg GTC, 0 mg CCA, and 119 mg caffeine (Table 1). GTC and CCA in the active beverage and PLA beverages were determined by high-performance liquid chromatography and dispensed industrially into bottles. Table 1 shows the polyphenols composition of the GTC and CCA in active and PLA beverages.

### 2.4. Analytical Procedure

Plasma and serum samples, collected from participants in the fasting state and after intake of the test meal and test beverages, were snap-frozen in liquid nitrogen and stored at −80 °C until the analyses. A plasma sample was used for glucose measurements, and a serum sample was used for insulin measurements. Samples for measurement of glucagon, active GLP-1, and total GIP were collected using BD P800 (Becton Dickenson, London, UK), and their levels were measured using ELISA kits (glucagon: Mercodia, Uppsala, Sweden; active GLP-1: Immune Biology Laboratories, Gunma, Japan; total GIP: Merck Millipore, Darmstadt, Germany). Coefficients of variation were intra-assay for glucagon: 2.3%, GLP-1: 5.3%, GIP: 9.6%; inter-assay for glucagon: 4.3%, GLP-1: 9.2%, GIP: 7.2%. All other indices were analyzed at LSI Medience, Co., Ltd. (Tokyo, Japan). Total AUCs for glucose, insulin, glucagon, C-peptide, GLP-1, and GIP were calculated using the trapezoidal rule. The Matsuda index of insulin sensitivity [25,26,27] was calculated from measurements obtained after the test meal consumption. The insulin and glucose measurements were used to calculate the HOMA-IR, HOMA-β, and AUC_ins×glu_ [28,29,30].

### 2.5. Statistical Analysis

Data were expressed as mean ± standard error unless otherwise indicated. The AUCs were calculated using the trapezoidal rule, maximum concentration (C_max_), and effect size. The 95% confidence intervals (CI) for the primary and secondary outcome measures were calculated from values measured at each time point. The sample size was estimated based on the results of a preliminary study as 12, assuming a significance level of 5% and a statistical power of 80%. In the preliminary study, the effect size of the glucose AUC was −0.681 (mg/dL·2 h) with a standard deviation of 0.857 (mg/dL·2 h). The following statistical analyses were performed for the GTC+CCA and PLA conditions. Regarding AUC and C_max_, differences between the first and second phases were determined for each subject, and comparisons were performed between the sequence group using an unpaired *t*-test to evaluate the effects of the interventions. Differences in the values at each time-point were determined for each participant and entered into a linear mixed-effect model with treatment, time, and treatment-by-time interaction as the fixed effects. An unstructured variance-covariance matrix was applied to compare time points, and empirical variances (EMPIRICAL option) were used to estimate the fixed effects. The significance of the fixed effects was evaluated, and comparisons between treatments at each time point were performed [31]. Pearson’s correlation coefficients or Spearman’s rank correlation coefficients analyzed the relationship between pairs of variables. To confirm the involvement of GLP-1 in altered insulin resistance due to GTC+CCA, we performed a post hoc analysis of correlations of the difference in insulin resistance with the difference in GLP-1 between GTC+CCA and PLA conditions. In all analyses, the significance level was 5%. Significant differences were tested using the following statistical analysis software: Microsoft Excel 14.0, SAS version 9.4 (SAS Institute Inc., Cary, NC, USA), and StatXact 8, CrossOver module (Cytel Software, Cambridge, MA, USA).

## 3. Results

### 3.1. Subjects

Eleven healthy men completed the trial, and all data collected were analyzed (Figure 1). The number of subjects (n = 11) assessed and enrolled in the study was one less than the estimated number of subjects needed (n = 12). The anthropometric measurements and blood glucose-related hormone levels at baseline are presented in Table 2. Based on three days of dietary records prior to the trial, the mean nutrient intake was: total energy intake for PLA: 2234 ± 85 kcal/d, GTC+CCA: 2219 ± 90 kcal/d; protein intake for PLA: 82.8 ± 3.0 g/d, GTC+CCA: 82.8 ± 4.4 g/d; fat intake for PLA: 74.5 ± 4.1 g/d, GTC+CCA: 71.6 ± 4.1 g/d; and CHO intake for PLA: 288.8 ± 10.5 g/d, GTC+CCA: 296.9 ± 11.4 g/d. Mean energy intake, protein, fat, and CHO during the last three days before the interventions did not differ significantly by treatment. Consumption of the GTC+CCA beverage for three weeks was associated with a slight but significant weight reduction compared to consumption of the PLA beverage (*p* = 0.035). No adverse events were observed.

### 3.2. Blood Glucose and Insulin

The levels of fasting and postprandial glucose were within the normal range in all subjects. Comparisons of measurements between the PLA and GTC+CCA conditions using linear mixed-effect models revealed a treatment by time interaction (*p* = 0.002) for the change in blood glucose but no main treatment effect (*p* = 0.183) (Figure 2). The total AUC of glucose did not differ significantly between conditions (PLA: 367 ± 13 mg/dL·4h; GTC+CCA: 350 ± 10 mg/dL·4h; effect size [95% CI]: −15.55 [−50.78 to 19.68]; *p* = 0.3442), however, the C_max_ for postprandial blood glucose was significantly lower in the GTC+ PLA condition (PLA: 118.5 ± 5.2 mg/dL; GTC+CCA: 105.5 ± 4.6 mg/dL; effect size [95% CI]: −12.57 [−24.84 to −0.29]; *p* = 0.046). Analysis of insulin changes under the GTC+CCA condition compared to the PLA condition also revealed the main treatment effect (*p* = 0.003) and a treatment-by-time interaction (*p* = 0.028). Moreover, the total AUC of postprandial insulin (PLA: 75.6 ± 9.3 μU/mL·4h; GTC+CCA: 57.5 ± 6.5 μU/mL·4h; effect size [95% CI]: −17.02 (−27.21 to −6.82); *p* = 0.004 ] and C_max_ [PLA: 41.5 ± 5.6 μU/mL; GTC+CCA: 30.6 ± 4.1 μU/mL; effect size [95% CI]: −9.88 [−18.18 to −1.58]; *p* = 0.025) were significantly lower under the GTC+CCA condition. The change in C-peptide revealed a main effect of treatment (*p* = 0.012) and treatment-by-time interaction (*p* = 0.044). Additionally, the change in glucagon revealed the main treatment effect (*p* = 0.004) and a treatment-by-time interaction (*p* = 0.001) under the GTC+CCA condition compared with the PLA condition.

### 3.3. Incretins

The change in the postprandial GLP-1 revealed the main treatment effect (*p* = 0.002) and a treatment-by-time interaction (*p* < 0.001) in a mixed-effect model of the measurement data (Figure 2). Moreover, consuming GTC+CCA for three weeks significantly increased the postprandial GLP-1 AUC compared with the PLA (PLA: 25.3 ± 3.2 pmol/L·4h; GTC+CCA: 37.2 ± 4.2 pmol/L·4h; effect size [95% CI]: 11.50 [4.88 to 18.13]; *p* = 0.003). The change in the postprandial GIP also revealed a main treatment effect (*p* = 0.005) and a treatment-by-time interaction (*p* = 0.003). Furthermore, the postprandial GIP AUC_4h_ was significantly decreased under the GTC+CCA condition compared with the PLA condition (PLA: 1810.3 ± 159.2 pg/mL·4h; GTC+CCA: 1383.2 ± 105.4 pg/mL·4h; effect size [95% CI]: −413.5 [−664.6 to −162.4]; *p* = 0.005).

### 3.4. Insulin Sensitivity and Insulin Secretion Indices

Statistical analyses of the HOMA-IR (index of fasting insulin resistance), AUC_ins×glu_ (index of postprandial insulin resistance), Matsuda index (index of insulin sensitivity), and HOMA-β (index of insulin secretion) are presented in Table 3. Consuming the GTC+CCA beverage led to a significant decrease in the AUC_ins×glu_ of postprandial insulin resistance (PLA: 8367 ± 1686; GTC+CCA: 5630 ± 1032; effect size [95% CI]: −2558 [−4697 to −420]; *p* = 0.024), and a significant increase in the Matsuda index of insulin sensitivity (PLA: 11.3 ± 1.5; GTC+CCA: 14.6 ± 1.4; effect size [95% CI]: 2.96 [0.88 to 5.05]; *p* = 0.011). The fasting insulin resistance and secretion capacity indices were not significantly altered by consuming the GTC+CCA beverage.

### 3.5. Blood Biochemical Values

The blood sample measurements are presented in Figure 2 and Table 3. Mixed-effect models of the measurement data revealed treatment by time interactions for postprandial free fatty acid (*p* = 0.002), 3-hydroxybutyric acid (*p* = 0.002), and cortisol (*p* = 0.006), and a main treatment effect (*p* < 0.001) and a treatment-by-time interaction (*p* = 0.003) for bile acid. None of the measured indices of fasting blood glucose control differed remarkably between both conditions, except that glycoalbumin decreased after GTC+CCA consumption in 64% of the subjects (*p* = 0.167).

### 3.6. Correlations between the Differences in Insulin Sensitivity with the Difference in GLP-1

Figure 3 illustrates the correlation analyses of the change in the GLP-1 AUC with the change in the AUC_ins×glu_ and the Matsuda index change between the GCT+CCA and PLA conditions. GLP-1 change positively correlated with a change in the Matsuda index of insulin sensitivity between the GTC+CCA and PLA conditions (r = 0.655, *p* = 0.029). Further, the differences in changes in the GLP-1 AUC between the GTC+CCA and PLA conditions tended to be negatively correlated with the differences in postprandial insulin resistance due to GTC+CCA consumption (r = −0.553, *p* = 0.078).

## 4. Discussion

In this study, the effects of the continuous consumption of combined GTC and CCA on glucose metabolism, insulin sensitivity, and incretin secretion were evaluated. To the best of our knowledge, this study is the first to report the effects of the combined consumption of GTC and CCA, and our findings suggest that their combined consumption significantly improved postprandial whole-body insulin sensitivity, markedly increased GLP-1 secretion, decreased GIP secretion, and enhanced insulin sensitivity, possibly by increasing GLP-1 secretion.

The present study revealed that combined supplementation with GTC and CCA significantly decreased insulin resistance index (AUC_ins×glu_) values and improved insulin sensitivity index (Matsuda index) values after the high-fat and high-carbohydrate meal. Previous intervention studies have demonstrated that consuming green tea or GTC improves insulin resistance [15,17,20,32,33]. In animal studies, GTC consumption leads to a significant decrease in extracellular lipids in muscle, which is one of the mechanisms for improving insulin resistance [34]. A review on coffee and CCA also reported that short-term consumption of coffee, in contrast to habitual coffee consumption, might improve insulin sensitivity [35,36]. In animal studies, CCA increases sodium-glucose co-transporter-1 levels and promotes GLP-1 secretion [37]. Regarding the enhanced GLP-1 secretion following GTC and CCA consumption (additionally or synergistically), the increase observed in this study was remarkable compared to that in a previous study of single polyphenol consumption (GCT or CCA alone) that evaluated the association between improved postprandial hyperglycemia due to coffee polyphenols alone and increased GLP-1 secretion [22], or a study that examined the effects of CCA independently on postprandial GLP-1 secretion [38]. These findings indicate that combined supplementation with dietary polyphenols from green tea and coffee may effectively prevent diabetes onset due to their synergistic beneficial effects compared to single-polyphenol supplementation.

Among other food components, a single ingestion of green plant membranes and their continuous consumption for 90 days increases postprandial GLP-1 secretion [39]. The direct effect of green plant membrane ingestion on intestinal endocrine cells prolonged the process for digestion and absorption of dietary fat and carbohydrates in the small intestine and had an indirect effect via neural signals, such as cholecystokinin; these are suggested mechanisms underlying the increase in GLP-1 secretion [39]. The weight loss we observed after three weeks of combined GTC and CCA consumption may be partly due to appetite suppression resulting from the increased postprandial GLP-1 secretion or inhibitory effects of suppressed GIP secretion on fat accumulation. Furthermore, changes in daily food intake during the intervention period should be considered. Administration of GLP-1 analogs significantly reduces body weight, mainly via changes in food intake [40,41]. Therefore, it is critical to understand the neural systems involved in the effects of GLP-1 analogs to reduce food intake and body weight [42,43]. In this study, the increase in postprandial GLP-1 might be related to the weight loss observed, and further evaluation is needed.

GLP-1 secretion induced by the combined consumption of GTC and CCA may be promoted by the following two mechanisms. First, the glucose-sensing mechanism via sodium-glucose co-transporter-1 or a GLP-1 exocytosis mechanism due to glucose metabolism in L cells might be directly affected by GTC and CCA, or by nutritional components altered by GTC and CCA [34,44,45]. Second, the suppressive effect of GTC and CCA on glucose absorption in the small intestine might change the site of absorption to the lower part of the small intestine, thereby increasing the stimulation of L cells [37,46,47]. However, the GLP secretion mechanism associated with the combined consumption of GTC and CCA remains to be clarified.

This study had limitations. First, the sample size was small, with eleven men, and the participants were healthy, non-diabetic individuals. Thus, it is unclear whether GLP-1 secretion would be similarly promoted or if insulin resistance would be improved in people with severe insulin resistance or diabetes. In addition, since there were no women in the study group, the results of this study can only be related to men and cannot be inferred in general. The effects of the combined consumption of GTC and CCA should be confirmed by additional studies with larger sample sizes, including men and women with insulin resistance. Moreover, caffeine is one of the pharmacologically active components of green tea and coffee. While the effect of caffeine consumption to improve glucose tolerance has been suggested, its effect on glucose levels varies from study to study [48]. In contrast, several studies have reported that caffeine increases insulin secretion but does not necessarily improve glucose levels in oral glucose challenge tests [49,50,51,52], suggesting that caffeine could influence insulin clearance. In this study, the caffeine quantity was the same under the PLA and GTC+CCA conditions; nonetheless, we cannot rule out the possibility that caffeine, GTC, and CCA contributed to the observed effects.

## 5. Conclusions

Our findings demonstrated that the continuous, combined consumption of GTC and CCA for three weeks suppressed hyperglycemia and insulin after consuming a high-fat test meal containing 75 g of glucose and improved insulin sensitivity in healthy males. In addition, combined consumption of GTC and CCA promoted postprandial GLP-1 secretion and suppressed GIP secretion, suggesting that the increase in GLP-1 might at least partly account for improvements in insulin sensitivity.

## Figures and Tables

**Figure 1 nutrients-14-05063-f001:**
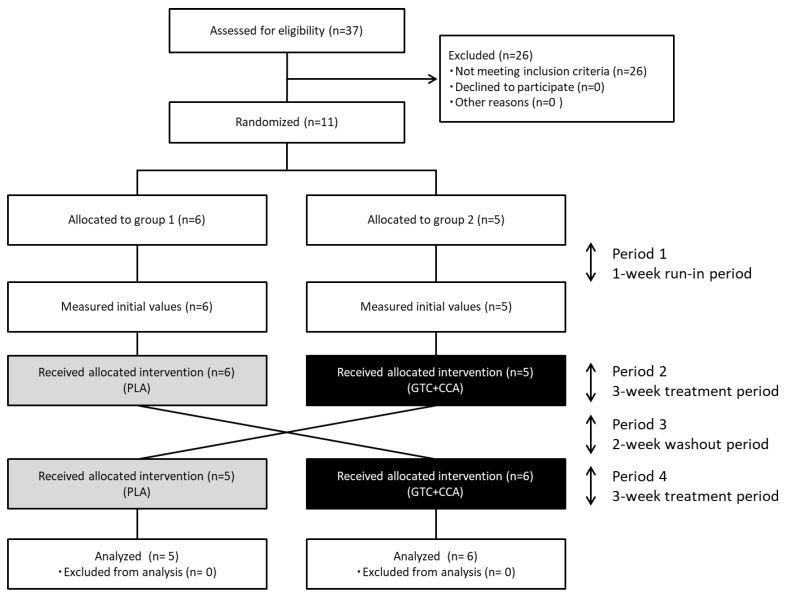
Diagram illustrating the flow of participants through each stage of the randomized crossover trial. Abbreviations: CCA, coffee chlorogenic acids; GTC, green tea catechins; PLA, placebo.

**Figure 2 nutrients-14-05063-f002:**
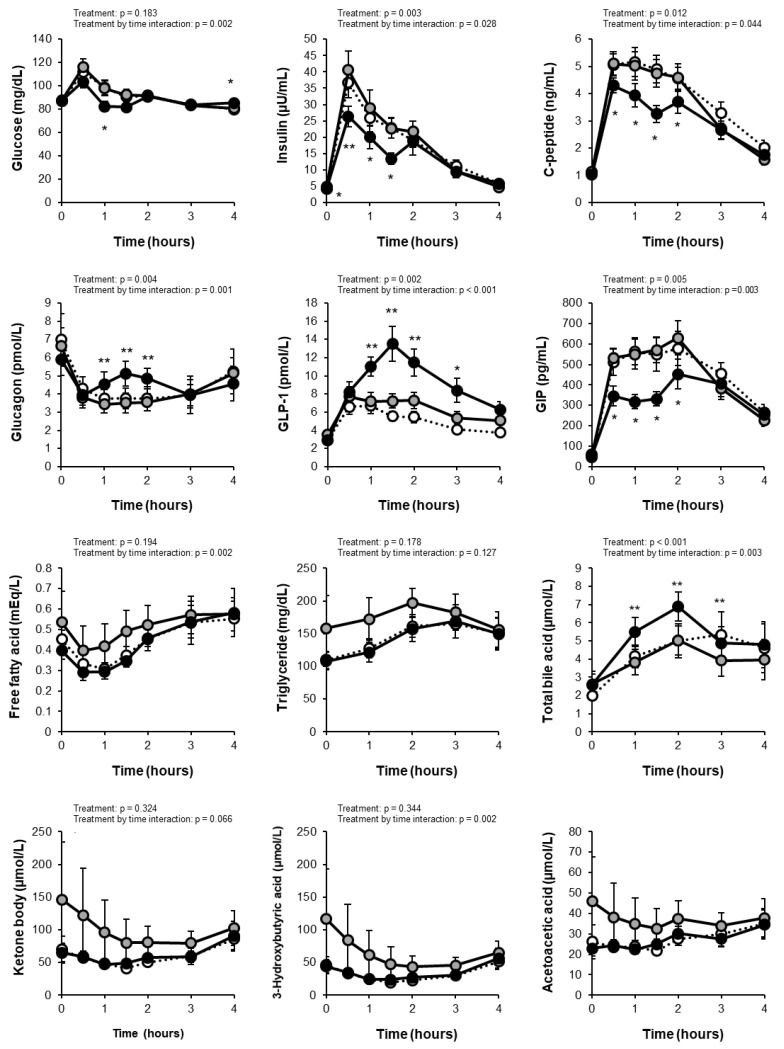
Changes in blood test results at fasting and after test meal consumption. White circles indicate the initial values. Gray and black circles indicate placebo and GTC+CCA, respectively. Treatment and treatment-by-time interactions were analyzed using linear mixed-effect models. Data are expressed as mean ± standard error. (n = 11) * *p* < 0.05, ** *p* < 0.01. The acetoacetic acid was not available for the statistical analysis due to a lack of convergence criteria. Abbreviations: AUC, area under the curve; CCA, coffee chlorogenic acids; GIP, glucose-dependent insulinotropic polypeptide; GLP-1, glucagon-like peptide-1; GTC, green tea catechins; PLA, placebo.

**Figure 3 nutrients-14-05063-f003:**
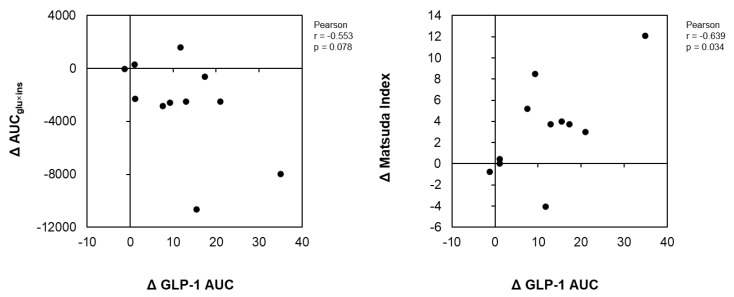
Correlations of differences in insulin resistance indices and AUC of GLP-1 after consuming GTC+CCA and PLA. ΔGLP-1, Δ AUC_glu×ins_, and ΔMatsuda index for each subject was calculated by subtracting the AUC of GLP-1, AUC_glu×ins_, and Matsuda index in the GTC+CCA treatment from those of the PLA treatment. The relationship between pairs of variables is presented using Pearson’s correlation coefficient (n = 11). Abbreviations: AUC, area under the curve; CCA, coffee chlorogenic acids; GLP-1, glucagon-like peptide-1; GTC, green tea catechins; PLA, placebo.

**Table 1 nutrients-14-05063-t001:** Composition of the test beverages.

		PLA	GTC+CCA
Catechin	mg	0	31
Epicatechin	mg	0	37
Gallocatechin	mg	0	134
Epigallocatechin	mg	0	120
Catechin gallate	mg	0	23
Epicatechin gallate	mg	0	41
Gallocatechin gallate	mg	0	108
Epigallocatechin gallate	mg	0	126
Total catechins	mg	0	620
3-caffeoylquinic acid	mg	0	104
4-caffeoylquinic acid	mg	0	94
5-caffeoylquinic acid	mg	0	102
3-feruloylquinic acid	mg	0	25
4-feruloylquinic acid	mg	0	23
5-feruloylquinic acid	mg	0	25
Total chlorogenic acids	mg	0	373
Caffeine	mg	119	119

Abbreviations: GTC+CCA, green tea catechin + chlorogenic acid; PLA, placebo.

**Table 2 nutrients-14-05063-t002:** Baseline physical characteristics, insulin-related indicators, and fasting blood constituents.

		Baseline
Age	y	41 ± 9
Height	cm	174.2 ± 4.0
Weight	kg	70.3 ± 9.4
Body mass index	kg/m^2^	23.1 ± 2.7
Body fat	%	20.3 ± 4.6
Waist circumference	cm	84.5 ± 8.7
Systolic blood pressure	mmHg	129 ± 15
Diastolic blood pressure	mmHg	75 ± 12
Body temperature	°C	36.1 ± 0.4
Glucose	mg/dL	87 ± 5
Insulin	µU/mL	4.2 ± 1.8
HOMA-IR		0.90 ± 0.36
Glucose AUC_2h_ × Insulin AUC_2h_		7366 ± 4021
Matsuda Index		12.3 ± 4.6
HOMA-β		67.3 ± 34.4
GA	%	13.3 ± 1.2
1,5-AG	µg/mL	24.3 ± 5.2
HbA_1c_	%	5.4 ± 0.3
h-CRP	mg/dL	0.036 ± 0.052
LDL-C	mg/dL	113 ± 24
HDL-C	mg/dL	56 ± 13
TC	mg/dL	193 ± 22
AST	U/L	19 ± 5
ALT	U/L	20 ± 11
ALP	U/L	170 ± 33
γ-GTP	U/L	28 ± 14
T4	µg/dL	7.1 ± 1.0
T3	ng/dL	104 ± 20
TSH	µIU/mL	1.337 ± 0.629

Data are expressed as mean ± standard deviation (n = 11). Abbreviations: HOMA-IR, homeostatic model assessment of insulin resistance index; GA, glycoalbumin; 1,5-AG, 1,5-anhydroglucitol; HbA1c, hemoglobin A1C; h-CRP, high-sensitivity C-reactive protein; LDL-C, low-density lipoprotein cholesterol; HDL-C, high-density lipoprotein cholesterol; TC, total cholesterol; AST, aspartate aminotransferase; ALT, alanine aminotransferase; ALP, alkaline phosphatase; γ-GTP, γ-glutamyl transpeptidase; T4, thyroxine; T3, triiodothyronine; TSH, thyroid-stimulating hormone.

**Table 3 nutrients-14-05063-t003:** Differences in physical characteristics, insulin-related indicators, and fasting blood constituents between conditions after the intervention.

		PLA	GTC+CCA	Effect of Difference * Mean (95% CI)	*p*-Value
Weight	kg	70.7 ± 2.7	70.1 ± 2.7	−0.52 (−0.99, −0.05)	0.035
Body mass index	kg/m^2^	23.2 ± 0.8	23.1 ± 0.8	−0.15 (−0.31, 0.01)	0.062
Body fat	%	20.4 ± 1.4	20.6 ± 1.6	0.28 (−0.47, 1.03)	0.422
Waist circumference	cm	85.0 ± 2.6	85.1 ± 2.6	0.15 (−0.66, 0.97)	0.681
Systolic blood pressure	mmHg	130.3 ± 4.2	124.8 ± 4.3	−5.32 (−15.91, 5.27)	0.285
Diastolic blood pressure	mmHg	80.3 ± 3.3	75.7 ± 3.7	−4.73 (−11.24, 1.77)	0.134
Body temperature	°C	36.3 ± 0.1	36.3 ± 0.1	0.02 (−0.15, 0.18)	0.823
HOMA-IR		1.07 ± 0.15	0.90 ± 0.08	−0.15 (−0.33, 0.03)	0.099
Glucose AUC_2h_ × Insulin AUC_2h_		8367 ± 1686	5630 ± 1032	−2558 (−4697, −420)	0.024
Matsuda Index		11.3 ± 1.5	14.6 ± 1.4	2.96 (0.88, 5.05)	0.011
HOMA-β		76.0 ± 10.3	64.6 ± 7.4	−9.50 (−22.08, 3.09)	0.122
GA	%	13.2 ± 0.4	13.0 ± 0.4	−0.19 (−0.48, 0.10)	0.167
1,5-AG	µg/mL	25.7 ± 1.7	24.9 ± 1.8	−0.79 (−1.84, 0.25)	0.121
HbA_1c_	%	5.3 ± 0.1	5.4 ± 0.1	0.04 (−0.03, 0.10)	0.245
h-CRP	mg/dL	0.04 ± 0.02	0.04 ± 0.01	−0.01 (−0.02, 0.01)	0.397
LDL-C	mg/dL	108.7 ± 7.0	110.9 ± 6.8	2.57 (−8.43, 13.56)	0.610
HDL-C	mg/dL	53.3 ± 2.7	52.1 ± 3.5	−1.30 (−4.41, 1.81)	0.369
TC	mg/dL	191.3 ± 7.5	186.9 ± 7.4	−3.48 (−20.55, 13.58)	0.655
AST	U/L	19.8 ± 1.6	18.9 ± 1.5	−0.97 (−3.39, 1.46)	0.390
ALT	U/L	17.3 ± 2.3	17.5 ± 2.2	0.28 (−3.48, 4.05)	0.869
ALP	U/L	176.2 ± 13.4	171.6 ± 10.5	−4.40 (−19.39, 10.59)	0.523
γ-GTP	U/L	26.8 ± 4.8	26.4 ± 4.3	−0.33 (−2.61, 1.94)	0.748
T4	µg/dL	7.1 ± 0.3	7.3 ± 0.3	0.18 (−0.46, 0.82)	0.543
T3	ng/dL	120.2 ± 7.1	121.2 ± 5.4	0.45 (−5.77, 6.67)	0.874
TSH	µIU/mL	1.3 ± 0.1	1.3 ± 0.1	0.01 (−0.22, 0.23)	0.942

Data are expressed as mean ± standard error. Differences between the PLA and GTC+CCA conditions were analyzed. * Difference means “CGA+CCA” treatment—“PLA” treatment (n = 11). Abbreviations: GTC-GCA, green tea catechin + chlorogenic acid; PLA, placebo; HOMA-IR, homeostatic model assessment of insulin resistance index; GA, glycoalbumin; 1,5-AG, 1,5-anhydroglucitol; HbA1c, hemoglobin A1C; h-CRP, high-sensitivity C-reactive protein; LDL-C, low-density lipoprotein cholesterol; HDL-C, high-density lipoprotein cholesterol; TC, total cholesterol; AST, aspartate aminotransferase; ALT, alanine aminotransferase; ALP, alkaline phosphatase; γ-GTP, γ-glutamyl transpeptidase; T4, thyroxine; T3, triiodothyronine; TSH, thyroid-stimulating hormone.

## Data Availability

The data used in this study are available from the corresponding author upon reasonable request.

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
