# Peer review of "Effects of Ingesting Both Catechins and Chlorogenic Acids on Glucose, Incretin, and Insulin Sensitivity in Healthy Men: A Randomized, Double-Blinded, Placebo-Controlled Crossover Trial"

_nutrients, 2022, doi:10.3390/nu14235063_

Round 1

Reviewer 1 Report

Nutrients

"Effects of Ingesting both Catechins and Chlorogenic Acids on Glucose, Incretin, and Insulin Sensitivity in Healthy Men: A Randomized, Double-blinded, Placebo-Controlled Crossover Trial"

Main observations:

The manuscript topic is consistent with the journal content.

LACK of Intra-assay coefficients of variation and inter-assay coefficients of variation for ELISA assay (glucagon, active GLP-1, total GIP).

In the paper, instead of HOMA-R - the popularly used abbreviation should be used: HOMA-IR- homeostatic model assessment of insulin resistance index.

 Due to the description: "Participants were asked to maintain their usual dietary habits and physical activities, and their intake of green tea and coffee was limited to once daily during the entire study period"; men in the placebo group were also allowed to use small amounts of active compounds (Catechins and Chlorogenic Acids).

Missing some units - for glucose and insulins in the table.

No information on whether glucose was measured in serum, plasma, or whole blood.

In Figure 2 - in the example of "Acetoacetid acid" - no information about  p-values (After Treatment and   After Treatment X Time )

In the case of the sentence: Further, the differences in changes in the GLP-1 AUC between the GTC+CCA and PLA conditions negatively correlated with the differences in postprandial insulin resistance due to GTC+CCA consumption (r = -0.553, p = 0.078); because p>0.05 - is at most a tendency for correlation to occur (the presence of correlation is out of line).

The discussion is consistent with the evidence and arguments and addresses the stated primary objective, but the literature is relatively outdated - more than 61 % are articles more than ten years old - more current items should be used.

LACK of LIMITATION of study (at the end of the Discussion section).

Among other things, it should include the following:

A small study group - of 11 men

Lack of women in the study - it is not possible to infer in general-just relative to men.

Minor observations:

The p-values on the graphs were presented by the authors in capital letters, and in the text, a small "p" was used - please standardize.

INSTEAD OF: The insulin and glucose measurements were used to calculate the HOMA-R, HOMA-β, and AUCins×glu [29-31].

SHOULD BE: The insulin and glucose measurements were used to calculate the HOMA-IR, HOMA-β, and AUCins×glu [29-31].

INSTEAD OF: Total areas under the curve (AUCs) for glucose, insulin, glucagon, c-peptide, GLP-1, and GIP were calculated using the trapezoidal rule.

SHOULD BE: Total areas under the curve (AUCs) for glucose, insulin, glucagon, C-peptide, GLP-1, and GIP were calculated using the trapezoidal rule.

Author Response

Response to Reviewer Comments

November 16, 2022

Nutrients

Dear Editors and reviewers

We would like to thank you very much for reviewing and commenting on our manuscript. We have carefully considered all your comments and made major revisions to the manuscript as a result. Below, we present our responses to each of the comments. We have significantly revised this manuscript in accordance with the comments. Please see the attached file. Our changes in the manuscript are indicated in track changes within the document. We believe that these changes have greatly improved our paper. We hope that our revised version satisfies the reviewers and editors.

-------------------------

Reviewer #1 comments:

Point 1: LACK of Intra-assay coefficients of variation and inter-assay coefficients of variation for ELISA assay (glucagon, active GLP-1, total GIP).

Response 1: Thank you for thoughtful suggestions. As you point out, coefficients of variation for ELISA assay should be considered. I have added text to the Methods section as you pointed out (P4 L165-166).

Point 2: In the paper, instead of HOMA-R - the popularly used abbreviation should be used: HOMA-IR- homeostatic model assessment of insulin resistance index.

Response 2: We appreciate your thoughtful review and helpful comments. We have implemented the appropriate corrections (P2 L82, P4 L171,P7 L221, P10 L283 P11 L296,

Point 3: Due to the description: "Participants were asked to maintain their usual dietary habits and physical activities, and their intake of green tea and coffee was limited to once daily during the entire study period"; men in the placebo group were also allowed to use small amounts of active compounds (Catechins and Chlorogenic Acids).

Response 3: We appreciate your thoughtful review. As you pointed out, the men in the placebo group were also not prohibited from taking active compounds. So they may have ingested small amounts of active compounds.

Point 4: Missing some units - for glucose and insulins in the table.

Response 4: We apologize very much for the inconvenience. It was our mistake and we have added the units to Table 2.

Point 5: No information on whether glucose was measured in serum, plasma, or whole blood.

Response 5: We appreciate your helpful comments. We have added text to the Methods section as you pointed out (P4 L160).

Point 6: In Figure 2 - in the example of "Acetoacetid acid" - no information about  p-values (After Treatment and   After Treatment X Time )

Response 6: Thank you for your kind review and helpful feedback. The statistical test for acetoacetic acid was not computable because the convergence criteria were not met. We have added text to the Figure legend (P9 L254-255).

Point 7: In the case of the sentence: Further, the differences in changes in the GLP-1 AUC between the GTC+CCA and PLA conditions negatively correlated with the differences in postprandial insulin resistance due to GTC+CCA consumption (r = -0.553, p = 0.078); because p>0.05 - is at most a tendency for correlation to occur (the presence of correlation is out of line).

Response 7: As you point out, we have corrected the text to the proper wording (P12 L319).

Point 8: The discussion is consistent with the evidence and arguments and addresses the stated primary objective, but the literature is relatively outdated - more than 61 % are articles more than ten years old - more current items should be used.

Response 8: We appreciate your helpful comments. As your suggestion, we have reviewed and improved our references (Ref 32, 33, 36, 41, 45, 52).

Point 9: LACK of LIMITATION of study (at the end of the Discussion section).

Among other things, it should include the following:

A small study group - of 11 men

Lack of women in the study - it is not possible to infer in general-just relative to men.

Response 9: We appreciate your helpful comments. Following the instructions, the conclusions section was placed independently, and the limitation section were clearly placed at the end of the discussion section. (P13 L391 and P13 L394).

Point 10: The p-values on the graphs were presented by the authors in capital letters, and in the text, a small "p" was used - please standardize.

Response 10: Thank you for pointing this out. We have unified the wording in Table 3.

Point 11: INSTEAD OF: The insulin and glucose measurements were used to calculate the HOMA-R, HOMA-β, and AUCins×glu [29-31].

SHOULD BE: The insulin and glucose measurements were used to calculate the HOMA-IR, HOMA-β, and AUCins×glu [29-31].

Response 11: We appreciate your suggestion and we have fixed it (P4 L171).

Point 12: INSTEAD OF: Total areas under the curve (AUCs) for glucose, insulin, glucagon, c-peptide, GLP-1, and GIP were calculated using the trapezoidal rule.

SHOULD BE: Total areas under the curve (AUCs) for glucose, insulin, glucagon, C-peptide, GLP-1, and GIP were calculated using the trapezoidal rule.

Response 12: As you suggested, we have fixed it (P4 L168).

-------------------------

Sincerely yours,

Aya Yanagimoto

Biological Science Research Labs.

Kao Corporation

2-1-3, Bunka, Sumida-ku,

Tokyo, 131-8501, Japan

Telephone: +81-3-5630-7476

Fax: +81-3-5630-7456

E-mail: yanagimoto.aya@kao.com

Reviewer 2 Report

It is my pleasure to review this paper entitled <Effects of Ingesting both Catechins and Chlorogenic Acids on Glucose, Incretin and Insulin Sensitivity in Healthy Men: A Randomized, Double-blinded, Placebo/Controlled Crossover Trial> by Yanagimoto A et al is article aims to give a point of view on effects of combined consumption of green tea catechins and coffee chlorogenic acids on postprandial glucose, the insulin incretin response, and insulin sensitivity. Authors should be congratulated for their work. The overall article is well written English is fluent and adequate, and the title is very informative about the paper. However, there are some drawbacks that could be addressed.

Please, the study population is too small to draw a conclusion, the authors need to enroll more patients.

The text is too long overall and the description is verbose.

I recommend making the text more compact and easier for the reader to understand the point.

Author Response

Response to Reviewer Comments

November 16, 2022

Nutrients

Dear Editors and reviewers

We would like to thank you very much for reviewing and commenting on our manuscript. We have carefully considered all your comments and made major revisions to the manuscript as a result. Below, we present our responses to each of the comments. We have significantly revised this manuscript in accordance with the comments. Please see the attached file. Our changes in the manuscript are indicated in track changes within the document. We believe that these changes have greatly improved our paper. We hope that our revised version satisfies the reviewers and editors.

-------------------------

Reviewer #2 comments:

It is my pleasure to review this paper entitled <Effects of Ingesting both Catechins and Chlorogenic Acids on Glucose, Incretin and Insulin Sensitivity in Healthy Men: A Randomized, Double-blinded, Placebo/Controlled Crossover Trial> by Yanagimoto A et al is article aims to give a point of view on effects of combined consumption of green tea catechins and coffee chlorogenic acids on postprandial glucose, the insulin incretin response, and insulin sensitivity. Authors should be congratulated for their work. The overall article is well written English is fluent and adequate, and the title is very informative about the paper. However, there are some drawbacks that could be addressed.

Point 1: Please, the study population is too small to draw a conclusion, the authors need to enroll more patients.

Response 1: We appreciate your helpful comments. We have added text to the discussion section as you pointed out as the small sample size was clarified in limitation section. (P13 L393-395).

Point 2: The text is too long overall and the description is verbose.

I recommend making the text more compact and easier for the reader to understand the point.

Response 2: We appreciate your helpful comments. As you point out, the length of the text and how it is written should be considered. We have reviewed and improved our manuscript.

-------------------------

Sincerely yours,

Aya Yanagimoto

Biological Science Research Labs.

Kao Corporation

2-1-3, Bunka, Sumida-ku,

Tokyo, 131-8501, Japan

Telephone: +81-3-5630-7476

Fax: +81-3-5630-7456

E-mail: yanagimoto.aya@kao.com
